# Weekly Variations in the Workload of Turkish National Youth Wrestlers: A Season of Complete Preparation

**DOI:** 10.3390/ijerph18073832

**Published:** 2021-04-06

**Authors:** Hadi Nobari, Rui Silva, Filipe Manuel Clemente, Zeki Akyildiz, Luca Paolo Ardigò, Jorge Pérez-Gómez

**Affiliations:** 1Department of Physical Education and Sports, University of Granada, 18010 Granada, Spain; 2Department of Exercise Physiology, Faculty of Sport Sciences, University of Isfahan, Isfahan 81746-7344, Iran; 3HEME Research Group, Faculty of Sport Sciences, University of Extremadura, 10003 Cáceres, Spain; jorgepg100@gmail.com; 4Escola Superior Desporto e Lazer, Instituto Politécnico de Viana do Castelo, Rua Escola Industrial e Comercial de Nun’Álvares, 4900-347 Viana do Castelo, Portugal; rui.s@ipvc.pt (R.S.); filipe.clemente5@gmail.com (F.M.C.); 5Instituto de Telecomunicações, Delegação da Covilhã, 1049-001 Lisboa, Portugal; 6Movement and Training Science Department, Gazi University, 06560 Ankara, Turkey; zekiakyldz@hotmail.com; 7Department of Neurosciences, Biomedicine and Movement Sciences, School of Exercise and Sport Science, University of Verona, 37129 Verona, Italy

**Keywords:** athlete monitoring, performance, training load, sports training, ACWR

## Abstract

The aim of this study was twofold: (1) to describe the weekly acute workload (wAW), chronic workload (wCW), acute/chronic workload ratio (wACWR), training monotony (wTM), and strain (wTS) across the preparation season (PS), and (2) to analyze the variations of wAW, wCW, wACWR, wTM, and training strain (wTS) between periods of PS (early-, mid-, and end). Ten elite young wrestlers were monitored daily during the 32 weeks of the season. Internal loads were monitored using session rating of perceived exertion, and weekly workload measures of wACWR, wTM, and wTS were also calculated. Results revealed that the greatest differences were found between early- and mid-PS for wAW (*p* = 0.004, *g* = 0.34), wCW (*p* = 0.002, *g* = 0.90), wTM (*p* = 0.005, *g* = 0.39), and wTS (*p* = 0.009, *g* = −1.1), respectively. The wACWR showed significant differences between early- and end-PS (*p* ≤ 0.001, *g* = −0.30). We concluded that wAW, wCW, and wTM are slightly lower during the first weeks of the PS. The wTM remained relatively high during the entire season, while wAW and wCW remained balanced throughout the PS. The greatest workload changes seem to happen from the early to mid-PS season.

## 1. Introduction

The systematic and continuously monitoring of training loads allows to control the dose-response of the training process which help coaches to analyze athlete’s daily variations during training and competition [1]. Moreover, it is a great tool for guaranteeing that the core principles of training such as individualization, variation and progressive overload are being followed [2]. Given that it is possible to know if the dose-response is being adequate allowing coaches to prevent athletes from the risks of overreaching and/or undertraining exposures [3].

Training load quantification is divided in two main categories [4]: (i) internal load (physiological and biological responses to a training stimulus) and (ii) external load (loads imposed by the exercise itself). The internal load is commonly assessed via heart rate monitor that allows to analyze some measurements from the heart rate, such as, maximal and resting heart rate, training impulse, and heart rate variability. Saliva concentrations, biochemical, hormonal, and immunological markers may also be used to assess internal load [5]. However, those methods are invasive and expensive. Furthermore, as wrestling is an intermittent sport with high-intensity bursts [6], the use of heart rate measurements may be compromised by the lower intensity tasks/actions during training and competition [7]. Despite this, other subjective methods for the assessment of internal loads are available. The use of rate of perceived exertion (RPE) scale proved to be valid and reliable [8,9]. Which recently is increasing use in teenage athletes [10,11,12,13]. In specific, a modification was made to obtain an indicator of internal load of an entire session (s-RPE), which is the multiplication of a given RPE score by the session time in minutes [3].

For this reasons, s-RPE is one of the most commonly internal load monitoring tools used in sports due to its practicability and ease of use [14,15]. Regarding combat sports, karate seems to be the sport with more research on s-RPE [16]. For instance, in a study conducted on 11 karate athletes from the Brazilian national team, it was found weekly s-RPE values of 2600 A.U. during one week of a training camp [17]. Moreover, in other study conducted on eight elite karatekas, it was found a mean of approximately 450 A.U. for a single training session [18]. On the other hand, in a taekwondo study of male and female elite athletes, it was found lower values (~250 A.U.) of s-RPE [19]. It was evidenced that the use of RPE measurements in combat sports is also an optimal and accurate tool for training and competition load quantification, for both young and adult athletes [16]. However, attention should be given to the fact that training loads and their respective workload indices may have different patterns between striking and grappling combat sports, since techniques and mechanical actions are different [16]. Since combat sports may vary in accordance with the intensity and volume of training, RPE should be an easy-to-use daily practice for regulation of the load imposed on the players and to control progression and variability.

Considering the limitations of using internal load measurement devices in a sport like wrestling, where it is neither allowed nor safe to use any type of accessory on the body, using s-RPE and associated workload indices, such as acute/chronic workload ratio (wACWR), training monotony (wTM) and training strain (wTS) is paramount for load monitoring [3]. In brief, the wACWR allows to analyze whether chronic loads are high enough to with-stand the acute loads imposed to athletes, and thus prevent from load spikes [20]. This approach may allow coaches to control the progression in the load in a week-to-week basis. The wTM refers to weekly load variations, and it is calculated by dividing the daily mean loads by their standard deviation, and strain refers to the tension imposed by loads and is the product of weekly loads and monotony. The wTM can be considered an important measure to control the variability of the load induced on the athletes.

Although there is considerable research regarding the topic of RPE and the associated workload indices in team sports, there is a lack of this type of research on combat sports, specifically in wrestling [16]. In fact, and to the best of our knowledge, studies investigating other workload indices beyond RPE and their variations across an entire season are lacking in wrestling youth athletes, with only one study, judo athletes, analyzing the weekly acute loads and wTS values, and their variations during a traditional periodized training season [21]. Therefore, the two objectives of this study were (1) to describe the weekly acute workload (wAW), chronic workload (wCW), acute and chronic workload ratio (wACWR), training monotony (wTM), and strain (wTS) across the preparation season (PS), and (2) to analyze the variations of wAW, wCW, wACWR, wTM, and wTS between periods of PS (early-, mid-, and end).

## 2. Materials and Methods

### 2.1. Participants

Ten national level young wrestlers participated in this study (mean ± standard deviation (SD); age, 16 ± 0.7 years; height, 163 ± 4.8 cm; body mass, 57.7 ± 9.0 kg; VO_2max_, 48.7 ± 1.4 mL.kg^−1^.min^−1^). As shown in Table 1.

Wrestlers participate in competitions organized by the National Turkish Wrestling Federation (NTWF). Inclusion criteria were included: (i) Wrestlers had to be training session in 90% of the PS to analyze the information; (ii) Wrestlers were not allowed participate in another training plan along the PS; (iii) Wrestlers had to be in the national team camp during the PS. Moreover, due to attending the camp, rest, sleep, nutrition, and temperature variations at the sports, camp center were the same for all participants throughout the PS. The training days are shown in Table 2, Table 3 and Table 4, respectively, based on the dominant microcycle of each period of the PS. The study was conducted in accordance with the Declaration of Helsinki, prior to the start players and their parents, signed informed consent to participate in this study, which was approved by the Ethics Committee of the Afyon Kocatepe University (ethical approval code number: NOM9).

### 2.2. Sample Size

According to statistical method analyzed, we estimated power and sample size for the design by F-test: within-group factor in a repeated measure. There is an 88.6% chance of correctly rejecting the null hypothesis of no difference in workload monitoring results across time with a total of 10 wrestlers.

### 2.3. Study Design

This study is a descriptive longitudinal for the entire season followed for the NTWF under 17 years. Daily viewing by players for 32 weeks from the start of the PS (Table 5). The first 11 weeks of the season were examined as the early-PS period, then 11 weeks as the mid-PS and finally 10 weeks as the end-PS. Players trained at least 5 times a week throughout the season. The players had been using the RPE questioners for the three year. They individually reported the RPE 30 min after the training session [3]. Then, the workload was calculated by multiplying the session (s-RPE) and the training time, for each training session. These data are weekly workload information reported in arbitrary units (AU) and analysis; wAW, wCW, wACWR, wTM, wTS [3].

### 2.4. Tests and Outcomes

Anthropometric: Anthropometric variables such as standing height (Seca model 654, Germany “with an accuracy of ± 5 mm) and weight (Seca model 654, with an accuracy of 0.1 per kg) were measured. These measurements were done in the morning [12,22]. The techniques considered by measurements were from the International Society for the Advancement of Kinanthropometry advanced [23]. Anthropometric measurements were repeated twice; the final score was recorded with an average of two measurements. If the technical error of anthropometric measurements had higher than 3%. Measurements were taken again and finally, the median of these three measurements was reported [13].

Aerobic Power Test: Intermittent Fitness Test 30–15 (30–15IFT) was performed to calculate the subjects’ maximum oxygen consumption (VO_2max_) [24]. However, this test cannot directly assess the VO_2max_. The 30–15IFT includes a 40-m shuttle and 15 s rest during 30 s of activity. The first stage was 30 s and the starting speed started at 8 km.h^−1^ and increased by 0.5 km.h^−1^ every 45 s. For all tests, subjects performed a standard dynamic warm-up of 15 min. After warming up, the subjects placed in groups of 5 stopped at the A line and after the loudspeaker sounded: Ready, Go! They started running to Lines B and C for 30 s. This test was continued until the subjects could not continue the test or the two-meter lines were not reached three times in a row. The subjects were encouraged to perform at their maximum performance during the test. Intra-class correlation coefficients in this study were calculated to test–retest reliability of 0.81 in this test.

Monitoring internal training loads: The internal loads of the athletes were determined using the s-RPE method. The 10-point Foster scale was used to monitoring the perceived effort of the players after each training session [25]. All ratings and RPE ratings of the athletes were made approximately 30 min after training. The training times of the athletes were calculated by multiplying the minutes with their RPE responses. Athletes were previously familiar with using RPE. They had been using RPE for at least three years. All athletes were informed about the RPE scale before the study started. Since it was thought that the players could be influenced by each other in the answers given to the RPE answers, the answers to the RPE questions of the players were taken individually from each player, multiplying the score in category-ratio scale (0 to 10 scale) by the duration of the session in minutes, as a measure of internal load.

Calculate training load: In this study, parameters workloads were calculated as follows: the wAW, which represents total of training load experienced in the previous seven days [8,12,26,27]; (2) the wCW, which represents the rolling exponential of average accumulated training load of training sessions experience in the previous three weeks [28]; (3) wACWR, which represents the used uncoupled formula [29], wACWR for week 5 equal to the wAW5/0.333 × (wCW in the previous three weeks); (4) wTM, equal to average wAW/SD; and (5) wTS, equal to wAW × wTM, both in one week [8]. All variables were calculated in each week of the experimental period.

### 2.5. Statistical Analysis

Statistical methods and calculations were performed using SPSS (version 22.0; IBM SPSS Inc., Chicago, IL, USA). Data are presented as mean and SD. Shapiro–Wilk and Levene’s tests were performed to check the normality and homogeneity of the information, respectively. Afterward, inferential experiments were executed. Variations of differences between the three in-PS periods were determined using a repeated-measures analysis of variance (ANOVA) with the average mean of the variables and then pairwise comparisons performed using the Bonferroni post hoc test. Partial eta squared (ηp^2^) was calculated as effect size of the repeated-measures ANOVA. Hedge’s g effect size with 95% confidence interval were also calculated to determine the magnitude of pairwise comparisons for both between-period comparatives [29,30,31]. The Hopkins’ thresholds for effect size statistics were used, as follow [32]: trivial (<0.2), small (≥0.2), moderate (≥0.5) and large (≥0.8) [30]. Significance level was set at *p* ≤ 0.05. We performed to calculate an a-priori estimation of power and sample size, the statistical software (G-Power; University of Dusseldorf, Dusseldorf, Germany) was applied. The selected study design: F-test; ANOVA, Repeated Measures; Within Factors; Power α err probability of 0.05, and Power 1-β err probability of 0.80.

## 3. Results

As demonstrated in Figure 1, the wAW, wCW and wACWR variations across the full PS and their different periods. The highest and lowest workloads were wAW (week (W)11 = 2992 ± 583.7 and W6 = 741 ± 210.1 AU), wCW (W13 = 2700.6 ± 165.3 and W8 = 1868.2 ± 291.4 AU), and wACWR (W7 = 1.49 ± 0.53 and W6 = 0.32 ± 0.09 AU), respectively.

As illustrated in Figure 2, the wTM and wTS variations across the full PS and their different periods. The highest and lowest workloads were wTM (W11 = 2.89 ± 1.80 and W6 = 0.49 ± 0.18 AU) and wTS (W11 = 9159.75 ± 7794.16 and W6 = 408.98 ± 122.34 AU), respectively.

The results of comparative differences between three periods in the Table 6 has been displayed for wAW, wCW wACWR, wTM, and wTS. The analysis showed in wAW (*p* = 0.04, *F* = 4.03, η_p_^2^ = 0.31), wCW (*p* = 0.002, *F* = 8.71, η_p_^2^ = 0.49), wACWR (*p* ≤ 0.001, *F* = 17.42, η_p_^2^ = 0.66), wTM (*p* = 0.05, *F* = 4.79, η_p_^2^ = 0.35), and wTS (*p* = 0.008, *F* = 9.45, η_p_^2^ = 0.70). We observed, mid-PS proffer a significant greater wAW (*p* = 0.004, *g* = 0.34), wCW (*p* = 0.002, *g* = 0.90), wTM (*p* = 0.005, *g* = 0.39), and wTS (*p* = 0.009, *g* = −1.1) compared to early-PS. Plus, the early-PS showed a significantly greater than wACWR compared to mid-PS (*p* = 0.008, *g* = −0.25) and end-PS (*p* ≤ 0.001, *g* = −0.30).

## 4. Discussion

The present study aimed (a) to describe the wAW, wCW, wACWR, wTM, and wTS across the PS, and (b) to analyze their variations between early-, mid-, and end-PS. The findings revealed that there was only one wAW spike during early-PS, and ACWR values remained in the safe zone across the PS. However, wTM remained above the recommended values during the PS. Regarding the second aim, results revealed that the greatest differences were found between early- and mid-PS for all variables.

Considering the weekly patterns of wAW, wCW, wACWR, it was clearly observable that the AW remained above 2000 A.U. per week. Only one week of taper (w6) followed by a load spike (w7) was observed in the entire season. The CW remained above 2000 A.U. throughout the season, except between weeks 6 to 8 in which CW remained with lower A.U. As the values of AW and CW remained balanced throughout the season, the values of wACWR remained in the “sweet spot” zone. However, the taper week (w6) caused an wACWR value of 0.3 A.U., which is below the recommended lower threshold (0.8 A.U.), and the following load spike (w7) a value of 1.5 A.U., which is above the higher threshold (1.3 A.U.). A study conducted on 10 young judo athletes, analyzed among others, the wAW and strain variations across a traditional periodized season [21]. Results of that study revealed that in preparation training blocks, the wAW maintained above 1500 A.U. and up to 2500 A.U., which is coincident with our findings.

The present study strain values revealed to be slightly lower during the early-PS, however, wTS values remained between ~4600 and 5500 A.U. during the entire season. Furthermore, TM showed high values (>2.0 A.U. threshold), in the overall weeks. Despite these presented values, attention should be given to the fact that there were high weekly coefficients of variation in most weeks, for all analyzed variables. In contrast with our results, the above mentioned study [21], showed wTS values between ~1500 and 2500 A.U. during preparation blocks. Values between 4600 and slightly above 5500 A.U. were observed in the present study during the entire season. The maintenance of training strain values observed throughout the season may be a result of the lack of weekly training load variation, given the high monotony values. Coaches should be aware of these load patterns and promote recovery sessions or “easy” days allowing training adaptations and preventing from overreaching [33,34]. Furthermore, although there was only one wAW spike during the entire season, it is imperative that coaches acknowledge the fact that weekly load spikes above 10% may be harmful for athletes’ performance and health [33]. However, depending on athletes and/or the different sport contexts, weekly load spikes up to ~25% may be tolerated [35].

Regarding the second purpose of the present study, results revealed significant increases of wAW and wCW from early- and mid-PS, while wACWR values significantly decreased from early- to mid-PS. The wTM values had significant increases from early- to end-PS, maintaining relatively high values across the season. In the study of Agostinho et al. [21], the acute and strain values were significantly greater in the first weeks of the season (preparation period meso-cycle) compared to the following five meso-cycles of training. This is contrary to our findings that revealed lower values of acute loads at the beginning of the season. These variations between periods of an entire season can vary according to the different methodologies and training ideas of wrestling coaches, as well as the different methods used in the studies that analyze workload variations [9,36,37]. In addition, competitive periods must be carefully planned, as a wrestling athlete may have to compete in up to six matches of 6 to 8 min of duration within a 48 h window, depending on the tournament [6].

It is important to mention that there is a lack of longitudinal studies regarding the different weekly workload profiles of grappling combat sports athletes that allow for a fair comparison with our results. The comparison with judo studies seemed to be more appropriate given the similarities with wrestling techniques. However, other combat sports studies [17,38], have investigated the responses of RPE although they did not consider other workloads such as wACWR, TM, TS, and their variations.

Given that, to the best of our knowledge, this is the first study reporting the wAW, wCW, wACWR, wTM, and wTS and their variations during a wrestling season of training. However, the present study was not without its limitations. Future studies should include a greater sample size for an increased generalizability of the presented evidence. We did not consider analyzing any external load measure. Due to the limitations of wrestling in using devices in training and competition clothes, it is impractical to use any external load measuring device and it may not be relevant to measure neither horizontal nor vertical dislocations in this sport. However, analyzing the duration of training sessions and competitions it would be of interest for future studies. Finally, we are aware about the debate currently on-going in the literature regarding the validity of the wACWR model for injury prevention purpose. Nevertheless, such a model is still considered valid by several researchers and scientific community has not reached a common agreement on its (lack of) validity, yet. Therefore, waiting for an eventual future acknowledged agreement and in presence of solid results such as within present study we chose to apply the model.

As practical implications it should be argued the need to employing training load monitoring processes, namely, using workload measures that allow control intra- and inter-individual variations in load and also understand the dynamics of load progression and variability. Using the current data, is also possible to provide descriptive information for future comparisons.

## 5. Conclusions

The first purpose of the present study was to describe the weekly workloads and their indices across the PS. Results revealed that although the wAW and wCW maintained balanced, the wTM values remained high during the entire PS. The aim of analyzing their variations between periods, revealed that the all workload parameters had significant changes between early- and mid-PS.

Our results give wrestling coaches new insights about the profiling of internal workload measures and their variations during an entire PS. Thus, attention should be given to the lower weekly load variations (high wTM values) and adjusting training accordingly.

## Figures and Tables

**Figure 1 ijerph-18-03832-f001:**
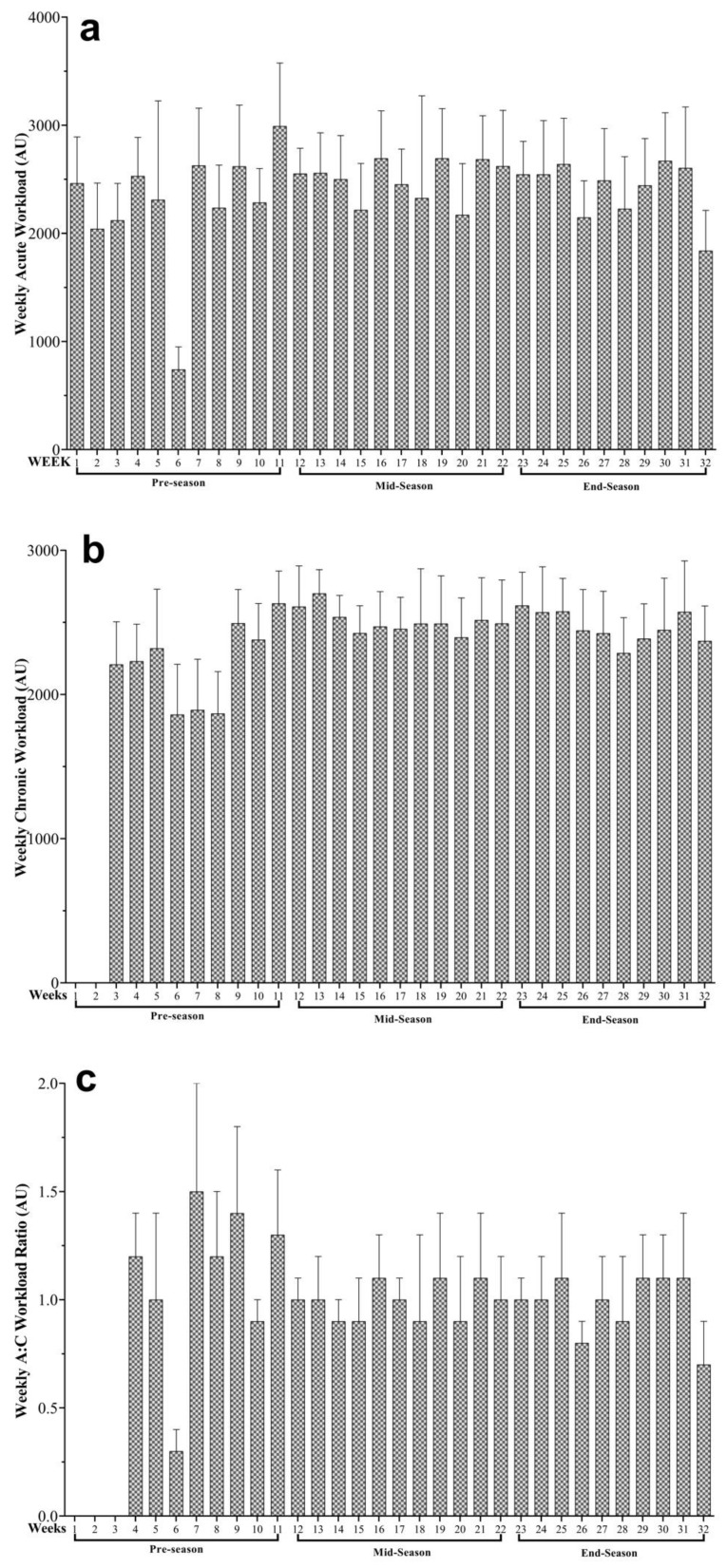
Descriptive statistics of (**a**) weekly average acute workload, (**b**) weekly average chronic workload, and (**c**) weekly average acute to chronic (A:C) workload ratio and their variations across the preparation season be shown in three periods (pre-, mid-, and end- season). Arbitrary units (AU).

**Figure 2 ijerph-18-03832-f002:**
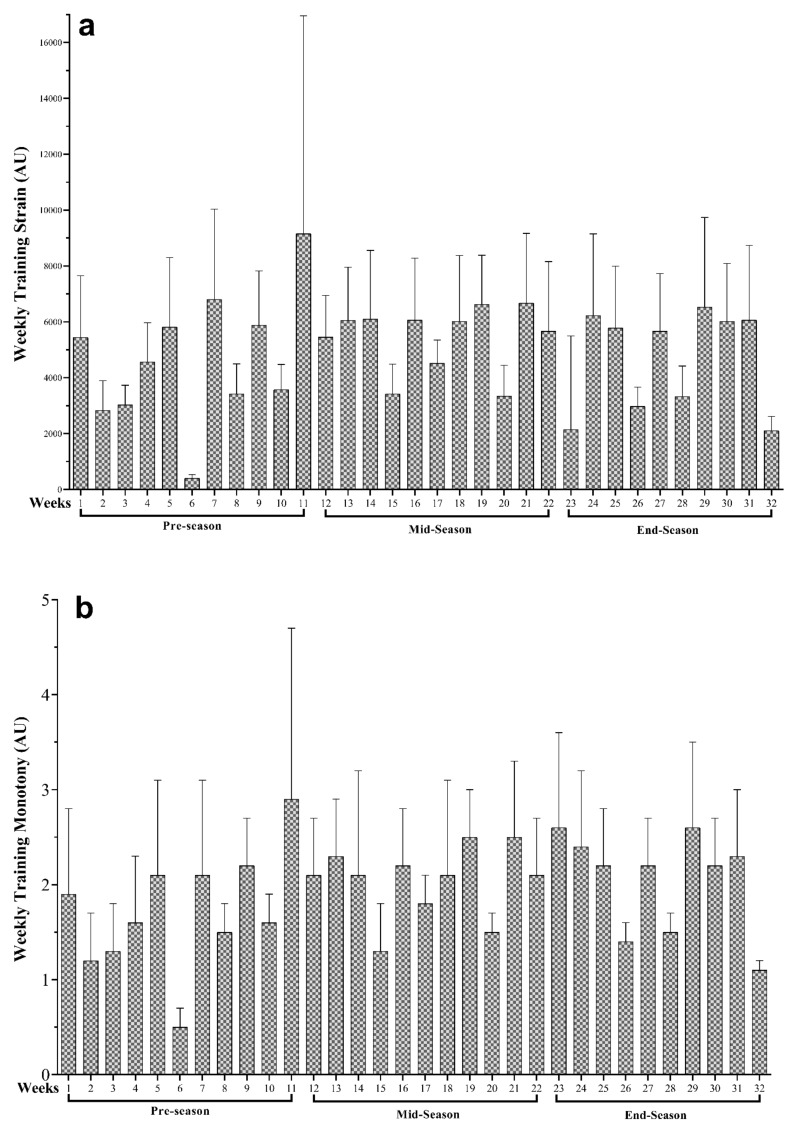
Descriptive statistics of (**a**) weekly average training strain and (**b**) weekly average training monotony and their variations across the preparation season be shown in three periods (pre-, mid- and end-season). Arbitrary units (AU).

**Table 1 ijerph-18-03832-t001:** Descriptive characteristics of the subjects.

Variables	Mean ± SD	Confidence Interval 95%
Height (cm)	163.0 ± 4.8	(162.7 to 163.3)
Body mass (kg)	57.7 ± 9.0	(52.1 to 63.3)
VO_2max_ (ml.kg^−1^.min^−1^)	48.7 ± 1.4	(47.8 to 49.6)
Age (years)	16.0 ± 0.7	(15.6 to 16.4)

SD: Standard deviation; VO_2max_: maximum oxygen consumption.

**Table 2 ijerph-18-03832-t002:** Design of a microcycle in the pre of the preparation season.

Days	Morning	Evening
Saturday	Light exercise—flexibility + complementary movements (pulling—rope) General structure I. Starting Practice: Flexibility and Conditioning Exercises (10 to 15 min) A. Neck circles and four-way neck exercises; B. Arm circles; C. Wrist and ankle circles; D. Belly circles; E. Leg stretches; F. Ankle circles; G. Bridging (side to side and backward and forward); H. Push-ups; I. Run and front roll intervals II. Wrestling Drill Work (15 min) A. Penetration drill; B. Push-pull drill; C. Spin drill to snap-down drill; D. Hip-heist drill * General warm-up in all wrestling training. Aerobic Training 30 min of low-intensity cardio. Heart rate in the 140–150 range	Weight training with maximum strength method (For heavyweights) and (For lightweight) 1. Bench Press 3 × 12^−15^ 2. Deadlift 3 × 12^−15^ 3. Overhead Barbell Press: 3 sets × 8^−10^ reps 4. Weighted Decline Sit-up: 3 sets × 20 reps 5. Hanging Leg/knee raise: 3 sets × 10^−15^ reps 6. Plank: 3 sets × 60 s
Sunday	Paired strength exercises that mimic wrestling moves (partner drills). Review of the technique with emphasis on general and local endurance 8–10 min in total (40–45 min)—around the anaerobic threshold Wrestling Drill Work A. Penetration drill; B. Push–pull drill; C. Spin drill to snap-down drill; D. Hip-heist drill	Combat practice with times longer than the actual match time in total (20–25 min) Teach New Move or Review Move A. Use step-by-step analysis of moves so wrestlers understand why and how they work. 1. Fireman’s carry instruction 2. Standing Peterson roll
Monday	OFF	Upper Body Plyometric Workout Plyometric Pushup: 3 × 5−10 Overhead Throw: 3 × 5−10 Medicine Ball slam: 3 × 5−10 Squat throws: 3 × 5^−10^ High İntensity Cardio Light jog for 5 min 5 × 50 m sprints (rest 2–3 min in between sprints)
Tuesday	Weight training with maximum strength method for (heavyweights) and for (lightweight) 1. Bentover Row 3 × 12 2. Bench Press 3 × 12 3. Squat 3 × 12 4. Rope climbing 7 m rope × 5 max speed 5. Romanian Deadlift 3 × 12	Wrestling Workout Session A. Neutral position (60% of wrestling workout session) B. Starting in referee’s position: offensive and defensive position (40% of wrestling workout session) Paired strength exercises that mimic wrestling moves (partner drills). A. Sprawl drill; B. Ankle–waist drill on whistle; C. Spin drill to snap-down drill; D. Stand-up (hand control) drill Finishing Practice: Conditioning Exercises (10 to 15 min) A. Run for 10 min (sprint and jog intervals) or jump rope; B. Strength exercises (such as sit-ups, push-ups, pull-ups on bar); C. Chalk talk as wrestlers cool down
Wednesday	Fighting practice With longer times than real race time Wrestling Workout Session (30 min) A. Neutral position (60% of wrestling workout session); B. Starting in referee’s position: offensive and defensive position (40% of wrestling workout session) A. Sprawl drill; B. Ankle–waist drill on whistle; C. Spin drill to snap-down drill; D. Stand-up (hand control) drill	Lower Body Plyometrics and High-Intensity Cardio 1. Squat Jumps: 3 × 5 × 10 2. Side to Side Lateral Hop: 3 × 5 × 10 3. Standing Long Jump: 3 × 5^−10^ High-Intensity Cardio – Light jog for 5 min – 5 × 50-m sprints (Rest 2–3 min between sprints)
Thursday	Aerobic Exercise In the range of 4–2 heart rate (Extensive Endurance)	Wrestling Drill Work A. Sprawl drill; B. Ankle–waist drill on whistle; C. Spin drill to snap-down drill; D. Stand-up (hand control) drill Wrestling Workout Session A. Neutral position (80% of wrestling workout session); B. Starting in referee’s position: offensive and defensive position (70% of wrestling workout session)
Friday	Anaerobic Workout (4 sets in total) 20 burpee; 20 horizontal jump; 20 jumping jack; 50 m run; 20 burpee; 20 long jump; 20 push up; 50 m run. There is no rest between movements. 2–3 min rest after all the movements	OFF Pool and sauna

**Table 3 ijerph-18-03832-t003:** Design of a microcycle in the mid of the preparation season.

Days	Morning	Evening
Saturday	Light exercise—flexibility + complementary movements (pulling—rope) General structure I. Starting Practice: Flexibility and Conditioning Exercises (10 to 15 min) A. Neck circles and four-way neck exercises; B. Arm circles; C. Wrist and ankle circles; D. Belly circles; E. Leg stretches; F. Ankle circles; G. Bridging (side to side and backward and forward); H. Push-ups; I. Run and front roll intervals II. Wrestling Drill Work (15 min) A. Penetration drill; B. Push-pull drill; C. Spin drill to snap-down drill; D. Hip-heist drill * General warm-up in all wrestling training. Anaerobic Training 5 × 400 m; 4 × 200 m; 2 ×100 m	Hang Clean: 7 × 1 Box Jumps: 7 × 3 Shuffle Pushups: 3 × 20 300 m Shuttle Lateral Wall Walks 3 × 15 m BB Walking Lunge Snatch Pull & Shrug: 3 × 4 Pistol Squats: 3 × 6 Floor Bench: 3 × 6 Lat Pulldowns: 3 × 6 Seated Rows: 3 × 5 Push Press: 3 × 3
Sunday	Review of the technique with emphasis on general and local endurance 8–10 min in total (40–45 min)—around the anaerobic threshold Wrestling Drill Work A. Penetration drill; B. Push–pull drill; C. Spin drill to snap-down drill; D. Hip-heist drill Paired strength exercises that mimic wrestling moves (partner drills).	Combat practice with times longer than the actual match time in total (20–25 min) A. Neutral position (60% of time); B. Starting in referee’s position: offensive and defensive position (70% of time)
Monday	OFF	Power workout Plyometric Push up: 3 × 10; Plyometric Box Jump: 3 × 5–8 Hang Clean: 3 × 3; Split Jerk: 3 × 4; Shrugs: 3 × 6; Push Jerk: 3 × 5
Tuesday	Weight training with maximum strength method For (heavyweights) and for (lightweight) 1. Bentover Row 3 × 12 2. Bench Press 3 × 12 3. Squat 3 × 12 4. Rope climbing 7 m rope × 5 max speed 5. Romanian Deadlift 3 × 12	Wrestling Workout Session A. Neutral position (60% of wrestling workout session); B. Starting in referee’s position: offensive and defensive position (40% of wrestling workout session) Standing shooting moves with 70% intensity. Finishing Practice: Conditioning Exercises (10 to 15 min) A. Run for 10 min (sprint and jog intervals) or jump rope; B. Strength exercises (such as sit-ups, push-ups, pull-ups on bar); C. Chalk talk as wrestlers cool down
Wednesday	Fighting practice With longer times than real race time Wrestling Workout Session (30 min) A. Neutral position (60% of wrestling workout session); B. Starting in referee’s position: offensive and defensive position (40% of wrestling workout session) Movements made on the ground (80%).	Clean Pulls + Shrugs: 4 × 2–6 Lat Pulldowns: 3 × 6 Seated Rows: 3 × 5 Split Jerk: 3 × 4 Wave Pushups: 3 × 5 Kettlebell Lunges: 3 × 10 Floor Bench: 3 × 6 Kettlebell Swings: 4 × 10
Thursday	Anaerobic Exercise (5 sets in total) Horizontal Jump; Clean Pulls; 50 m run; Kettlebell Swings; Lat Pulldowns; Split Jerk; 50 m run; Kettlebell Swings; Wave Pushups; Kettlebell Lunges; 50 m run; Kettlebell Swings. There is no rest between movements. 2–3 min rest after all the movements.	Wrestling Drill Work A. Sprawl drill B; Ankle–waist drill on whistle; C. Spin drill to snap-down drill D; Stand-up (hand control) drill Wrestling Workout Session A. Neutral position (80% of wrestling workout session); B. Starting in referee’s position: offensive and defensive position (70% of wrestling workout session) Both the ground and standing movements are 70% intensity.
Friday	Aerobic Exercise In the range of 4–2 heart rate (Extensive Endurance)	OFF Pool and sauna

**Table 4 ijerph-18-03832-t004:** Design of a microcycle in the end of the preparation season.

Days	Morning	Evening
Saturday	Light exercise—flexibility + complementary movements (pulling—rope) General structure I. Starting Practice: Flexibility and Conditioning Exercises (10 to 15 min) A. Neck circles and four-way neck exercises; B. Arm circles; C. Wrist and ankle circles; D. Belly circles; E. Leg stretches; F. Ankle circles; G. Bridging (side to side and backward and forward); H. Push-ups; I. Run and front roll intervals II. Wrestling Drill Work (15 min) A. Penetration drill; B. Push–pull drill; C. Spin drill to snap-down drill; D. Hip-heist drill * General warm-up in all wrestling training. Aerobic Training In the heart rate range of 140–150 Between 45 min and 60 min	Wave Squat: 3 × 8 Front Lunges: 3 × 6 Shuffle Pushups: 3 × 20 Kneeling Leg Curls: 3 × 8 Pistol Squats: 3 × 6 Floor Bench: 3 × 6 Lat Pulldowns: 3 × 6 Seated Rows: 3 × 5 Push Press: 3 × 3
Sunday	OFF	
Monday	OFF	Combat practice with times longer than the actual match time in total (20–25 min) A. Neutral position (40% of time); B. Starting in referee’s position: offensive and defensive position (40% of time); C. Repetition of light standing movements. Both the ground and standing movements are 60% intensity.
Tuesday	OFF	Weight training with maximum strength method for (heavyweights) and for (lightweight) Bentover Row 3 × 8−10 Bench Press 3 × 8−10 Squat 3 × 8−10 Incline Bench Press 3 × 8−10 Romanian Deadlift 3 × 12 Upright Rows 3 × 8−10
Wednesday	Fighting practice With longer times than real race time Wrestling Workout Session (30 min) A. Neutral position (60% of wrestling workout session); B. Starting in referee’s position: offensive and defensive position (40% of wrestling workout session) Focusing on the wrong techniques of athletes. Correction of faulty techniques.	Clean Pulls + Shrugs: 4 × 2−6 Lat Pulldowns 3 × 6 Seated Rows: 3 × 5 Split Jerk: 3 × 4 Wave Pushups: 3 × 5 Kettlebell Lunges: 3 × 10 Floor Bench: 3 × 6 Kettlebell Swings: 4 × 10
Thursday	Anaerobic Exercises (5 sets in total) Horizontal Jump; Clean Pulls; 50 m run; Kettlebell Swings; Lat Pulldowns; Split Jerk; 50 m run Kettlebell Swings; Wave Pushups; Kettlebell Lunges; 50 m run; Kettlebell Swings There is no rest between all movements, with a 2–3 min rest after all the movements.	Wrestling Drill Work A. Sprawl drill; B. Ankle–waist drill on whistle; C. Spin drill to snap-down drill; D. Stand-up (hand control) drill Wrestling Workout Session A. Neutral position (60% of wrestling workout session); B. Starting in referee’s position: offensive and defensive position (70% of wrestling workout session) Understanding escape and defense logic drills.
Friday	OFF Evaluate the status of competitors	OFF Pool and sauna

**Table 5 ijerph-18-03832-t005:** During monitoring in full season.

W (n)	1	2	3	4	5	6	7	8	9	10	11	12	13	14	15	16	17	18	19	20	21	22	23	24	25	26	27	28	29	30	31	32
TS (n)	7	6	6	7	7	3	7	6	7	6	7	7	7	7	6	7	7	7	7	6	7	7	7	7	7	6	7	6	7	7	7	5
Months	July				August				September				October				November				December				January				February			
Periods	Early preparation season											Mid preparation season											End preparation season									

TS, training session; W, week.

**Table 6 ijerph-18-03832-t006:** Comparison of workload variables between preparation season courses.

Variables	Season Period	Comparative	Mean Difference (95% CI)	*p*	Hedge’s *g* (95% CI)
wAW (AU)	EarPS: 2269.6 (729.9)	EarPS vs. MidPS	228 (−356.2 to 812.2)	**0.004**	0.34 (−0.5 to 1.2), S
	MidPS: 2497.5 (490.4)	EarPS vs. EndPS	145 (−365.0 to 655.5)	0.566	0.25 (−0.6 to 1.1), S
	EndPS: 2414.8 (238.8)	MidPS vs. EndPS	−83 (−445.1 to 279.7)	>0.999	−0.20 (−1.1 to 0.7), S
wCW (AU)	EarPS: 2209.6 (359.2)	EarPS vs. MidPS	298 (9.6 to 586.8)	**0.002**	0.90 (−0.02 to 1.8), L
	MidPS: 2507.8 (244.3)	EarPS vs. EndPS	260 (−26.4 to 546.7)	0.075	0.79 (−0.1 to 1.7), M
	EndPS: 2469.7 (238.8)	MidPS vs. EndPS	−38 (−265.1 to 188.9)	>0.999	−0.15 (−1.0 to 0.7), T
wACWR (AU)	EarPS: 1.11 (0.48)	EarPS vs. MidPS	−0.10 (−0.46 to 0.25)	**0.008**	−0.25 (−1.13 to 0.63), S
	MidPS: 1.00 (0.24)	EarPS vs. EndPS	−0.12 (−0.47 to 0.23)	**≤0.001**	−0.30 (−1.18 to 0.58), S
	EndPS: 0.99 (0.23)	MidPS vs. EndPS	−0.02 (−0.24 to 0.20)	>0.999	0.08 (−0.96 to 0.80), T
wTM (AU)	EarPS: 1.73 (0.86)	EarPS vs. MidPS	0.34 (−0.41 to 1.08)	**0.005**	0.39 (−0.49 to 1.28), S
	MidPS: 2.06 (0.71)	EarPS vs. EndPS	0.32 (−0.44 to 1.09)	0.256	0.37 (−0.51 to 1.25), S
	EndPS: 2.05 (0.75)	MidPS vs. EndPS	−0.01 (−0.70 to 0.68)	>0.999	−0.01 (−0.89 to 0.86), T
wTS (AU)	EarPS: 4633.8 (799.9)	EarPS vs. MidPS	−818.2 (−1443.0 to −193.3)	**0.009**	−1.1(−2.1 to −0.2), L
	MidPS: 5452.0 (494.6)	EarPS vs. EndPS	−53.1 (−869.8 to 763.7)	>0.999	−0.1 (−0.9 to 0.8), T
	EndPS: 4686.9 (933.6)	MidPS vs. EndPS	765.1 (63.2 to 1467.0)	0.222	0.9 (0.02 to 1.9), L

AU, arbitrary units; CI, confidence interval; wAW, weekly average acute workload in AU; wCW, weekly average chronic workload in AU; wACWR, weekly average acute:chronic workload ratio in AU; wTM, weekly average training monotony in AU; wTS, weekly average training strain in AU; EarPS, early-preparation season period; MidPS, mid-preparation season period; EndPS, end-preparation season period; T, Trivial; S, Small; M, Moderate; L, Large; *p*, *p*-value at alpha level 0.05; Hedges’s g (95% CI), Hedges’s g effect size magnitude with 95% confidence interval. Significant differences (*p* ≤ 0.05) are highlighted in bold.

## Data Availability

The datasets used and/or analyzed during the current study are available from the corresponding author on reasonable request.

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
