# Peer review of "Weekly Variations in the Workload of Turkish National Youth Wrestlers: A Season of Complete Preparation"

_ijerph, 2021, doi:10.3390/ijerph18073832_

Round 1
Reviewer 1 Report
I would like to thank the authors for the effort to reply to all my comments. I think the manuscript has improved. I have no other comments to add
Author Response
Dear Reviewer, we thank you for your comments in the previous steps for the scientific promotion of this article.
Reviewer 2 Report
I read with interest the changes that the authors made from the previous submission but unfortunately I have some doubts about how the authors interpret and data and in particular they interpret effect size.
In my opinion, the interpretation is wrong or something in the data analysis needs to be revised.
The reported effect size confidence intervals for some variables are not significant while the authors interpret them as such.
please clarify this point.
Author Response
Dear Reviewer,
Authors: Thanks for your comment. As we explained in statistics section, we conducted an ANOVA to analyze if there were differences between the different period of the season. Then, as ANOVA revealed significant differences, Bonferroni post hoc were applied to check pairwise comparisons.
Respect to the use of Hedge’s g computations, we calculate Hedge’s g to avoid positive biased outcomes, since it is well-known that the formula for Cohen’s d, which is based on sample averages gives a biased estimate of the population effect size (Hedges and Olkin, 1985), especially for small samples (n < 20). Although the difference between Hedges’s g and Cohen’s d is very small, especially in sample sizes above 20 (Kline, 2004), it is preferable (and just as easy) to report Hedges’s g (Lakens, 2013). Thus, our sample include a total of 10 participants (> 20). Therefore, we decided by the calculation of Hedge's g based on the aforementioned arguments, and for being consistent in the effect size parameter used in all comparisons included into the manuscript. Also, we have specified the threshold of effect size inside the table to make it more understandable to readers.
Hedges, L. V., and Olkin, I. (1985). Statistical methods for meta-analysis. San Diego, CA: Academic Press
Kline, R. B. (2004). Beyond Significance Testing: Reforming Data Analysis Methods in Behavioral Research. Washington DC: American Psychological Association. doi: 10.1037/10693-000
Lakens, D. (2013). Calculating and reporting effect sizes to facilitate cumulative science: a practical primer for t-tests and ANOVAs. Frontiers in Psychology, 4. doi: 10.3389/fpsyg.2013.00863
Reviewer 3 Report
Overall, I think this study is unique and adds the very important area of athlete monitoring. a major need for this write up though is to better explain what they did to being about those load measures. I think with the periodization and programming information this paper would be fantastic, but without it is simply a very surface report demonstrating that wrestlers trained across several months. Please provide insight into , 1) what the plan was, and 2) what were the exercises, the sets, reps, practice plans etc. that brought about these reported responses ?
Also, birefly
46) it can be expensive - for invasive, can use saliva instead of blood
56) dont start paragraph with "in fact"
53) are there studies on youth with sRPE?
97) pi think instead of saying "elite", "national level" would be much more appropriate
Author Response
Overall, I think this study is unique and adds the very important area of athlete monitoring. a major need for this write up though is to better explain what they did to being about those load measures. I think with the periodization and programming information this paper would be fantastic, but without it is simply a very surface report demonstrating that wrestlers trained across several months. Please provide insight into , 1) what the plan was, and 2) what were the exercises, the sets, reps, practice plans etc. that brought about these reported responses ?
Dear Reviewer, thank you for your suggestion, we have added tables based on each period of the PS.
Also, birefly
46) it can be expensive - for invasive, can use saliva instead of blood
Dear Reviewer, thank you, we have change it.
56) dont start paragraph with "in fact"
Dear Reviewer, thank you for your comments, we changed it.
53) are there studies on youth with sRPE?
Dear Reviewer, we added the related studies that used this method.
97) pi think instead of saying "elite", "national level" would be much more appropriate
Dear Reviewer, done.
Round 2
Reviewer 3 Report
thank you to the authors for providing the added detail of the training . well done.
This manuscript is a resubmission of an earlier submission. The following is a list of the peer review reports and author responses from that submission.
Round 1
Reviewer 1 Report
The present study aim (i) to describe the weekly acute workload, chronic workload, acute/chronic workload ratio, training monotony, and strain across the preparation season, and (ii) to analyze the variations of those parameters, between the different periods of the preparation season (early-, mid-and end).
Globally the manuscript contains several points of interest but it needs careful and deep revision. Several parts need to be rewritten in accord with the most recent paper regarding training load monitoring. In particular, the use of the acute/chronic workload ratio should be applied only as a numerical parameter that gives information on training progress and not as an injury prevention factor or “safe zone” (sweet point) (Impellizzeri et al 2020). Furthermore, given that the present study does not report information regarding sports injuries, it would be more appropriate to avoid this speculation. Nevertheless, it is interesting to see how the acute/chronic workload ratio and the other parameters vary within the preparation phases.
Additionally, the introduction needs a more logical structure that helps the reader to understand the purpose of the study. The Materials and methods session contains many imprecisions (see below). In particular, the use of some calculations and terminology seem inappropriate. For example, the term acute workload should not be associated with the sum of the weekly s-RPE but with the rolling average of training load experienced in the previous 7 days (Murray et al. 2017); moreover, despite authors deep describe anthropometric and aerobic tests no data are available. In particular, should be highlighted that Fitness Test 30-15 does not directly measure the maximum oxygen consumption.
The results should be reviewed based on the previous point. It is also not clear why a week of unloading/tapering is carried out during the first phase of the season. This could affect the outcome of the results and a different division could help the effective understanding of the data. In any case, a deeper discussion of this point should be performed in the discussion.
A review from a native English speaker is required.
Specific comment
Abstract.
Page 1, Line 18-20: I would suggest being consistent with the abbreviation in the test. In some cases, it seems that the same parameter is associated with different abbreviations. for example, the difference between wACWR and ACWR is not clear. Please correct it through the text.
Page 1, Line 23-25: I would suggest reporting data avoiding the statement “safe zones”, especially because it is not measured in the present study.
.
Introduction.
Page 2, Line 53: I would suggest changing the words “rate of perceived effort with “rate of perceived exertion”.
Page 2, Line 58 to 67: in these sentences, the authors report some training load data measured in different sports. These data are difficult to compare because, in addition to measuring loads deriving from different sports, the authors compare weekly load data with data from individual sessions, complicating the understanding of the text. For this reason, I would suggest improving this paragraph.
Page 2, Line 67 to 69: I would suggest adding more information regarding the importance of RPE in combat sports. Please could you add any examples or data regarding previous studies?
Page 2, Line 77 to 80: This point is controversial, I cautious approach should be considered.
Methods.
Page 3 line 132: Please could you provide data regarding these measurements.
Page 3 line 140 to 141: Please note that Fitness test 30-15 does not provide a direct measure of Vo2max.
Page 4 lines 162 to 163: Please could you provide more information regarding the different calculations. I would suggest adding mathematical formulas to a better understanding. Additionally, I would suggest using a different term for the weekly training load (sum of training sessions) and acute workload (rolling average or exponential, of the last 7 days). Please also consider correcting the calculation accordantly, if not already done.
Author Response
The present study aim (i) to describe the weekly acute workload, chronic workload, acute/chronic workload ratio, training monotony, and strain across the preparation season, and (ii) to analyze the variations of those parameters, between the different periods of the preparation season (early-, mid-and end).
Globally the manuscript contains several points of interest but it needs careful and deep revision. Several parts need to be rewritten in accord with the most recent paper regarding training load monitoring. In particular, the use of the acute/chronic workload ratio should be applied only as a numerical parameter that gives information on training progress and not as an injury prevention factor or “safe zone” (sweet point) (Impellizzeri et al 2020). Furthermore, given that the present study does not report information regarding sports injuries, it would be more appropriate to avoid this speculation. Nevertheless, it is interesting to see how the acute/chronic workload ratio and the other parameters vary within the preparation phases.
AUTHORS: DEAR REVIEWER, WE THANK YOU FOR YOUR ATTENTION AND COMMENTS FOR THIS STUDY.
Additionally, the introduction needs a more logical structure that helps the reader to understand the purpose of the study. The Materials and methods session contains many imprecisions (see below). In particular, the use of some calculations and terminology seem inappropriate. For example, the term acute workload should not be associated with the sum of the weekly s-RPE but with the rolling average of training load experienced in the previous 7 days (Murray et al. 2017); moreover, despite authors deep describe anthropometric and aerobic tests no data are available. In particular, should be highlighted that Fitness Test 30-15 does not directly measure the maximum oxygen consumption.
The results should be reviewed based on the previous point. It is also not clear why a week of unloading/tapering is carried out during the first phase of the season. This could affect the outcome of the results and a different division could help the effective understanding of the data. In any case, a deeper discussion of this point should be performed in the discussion.
A review from a native English speaker is required.
AUTHORS: DEAR REVIEWER, INTRODUCTION, AND DISCUSSION WERE CHANGED BASED ON SPECIFIC COMMENTS. WE ALSO CORRECTED THE SECTIONS ON CALCULATION METHODS AND FORMULAS. FINALLY, WE DISPLAYED THIS INFORMATION IN TABLE 1.
Specific comment
Abstract.
Page 1, Line 18-20: I would suggest being consistent with the abbreviation in the test. In some cases, it seems that the same parameter is associated with different abbreviations. for example, the difference between wACWR and ACWR is not clear. Please correct it through the text.
AUTHORS: DEAR REVIEWER, THANK YOU. WE HAVE STANDARDIZED ACROSS THE ARTICLE.
Page 1, Line 23-25: I would suggest reporting data avoiding the statement “safe zones”, especially because it is not measured in the present study.
AUTHORS: DEAR REVIEWER, THANK YOU. WE HAVE REMOVED.
Introduction.
Page 2, Line 53: I would suggest changing the words “rate of perceived effort with “rate of perceived exertion”.
AUTHORS: DEAR REVIEWER, THANK YOU. IT WAS CHANGED ACCORDINGLY.
Page 2, Line 58 to 67: in these sentences, the authors report some training load data measured in different sports. These data are difficult to compare because, in addition to measuring loads deriving from different sports, the authors compare weekly load data with data from individual sessions, complicating the understanding of the text. For this reason, I would suggest improving this paragraph.
AUTHORS: DEAR REVIEWER, THANK YOU. WE HAVE NOW FOCUSED ON COMBAT SPORTS.
Page 2, Line 67 to 69: I would suggest adding more information regarding the importance of RPE in combat sports. Please could you add any examples or data regarding previous studies?
AUTHORS: DEAR REVIEWER, THANK YOU. WE HAVE DEVELOPED THE IDEA IN THE END OF THE PARAGRAPH.
Page 2, Line 77 to 80: This point is controversial, I cautious approach should be considered.
AUTHORS: DEAR REVIEWER, THANK YOU. WE HAVE REMOVED AND CHANGED.
Methods.
Page 3 line 132: Please could you provide data regarding these measurements.
AUTHORS: DEAR REVIEWER, WE SPECIFIED THEIR INFORMATION IN THE PARTICIPANTS SECTION.
Page 3 line 140 to 141: Please note that Fitness test 30-15 does not provide a direct measure of Vo2max.
AUTHORS: DEAR REVIEWER, THANK YOU, WE ADDED THIS SENTENCE.
Page 4 lines 162 to 163: Please could you provide more information regarding the different calculations. I would suggest adding mathematical formulas to a better understanding. Additionally, I would suggest using a different term for the weekly training load (sum of training sessions) and acute workload (rolling average or exponential, of the last 7 days). Please also consider correcting the calculation accordantly, if not already done.
AUTHORS: DEAR REVIEWER, THANK YOU, THANK YOU, WE CONSIDERED THESE SUGGESTIONS AND MODIFIED.
Reviewer 2 Report
This study aims to investigate weekly variations in the workload of Turkish national youth wrestlers during a season. I think that the article is not innovative in term of knowledge about the topic but adds information about internal training in wrestler population.
Here my specific comments to the authors.
Lines 69-71
Why are there these differences? I suggest the authors to explain more in deep this point. How may the wrestling activity be affected by these differences?
Lines 78-80
I think that the relationship between acute/chronic workload ratio and injury is a lot discussed topic in the literature. I suggest of being more careful with this statement.
Lines 100-110
I think that the authors should describe more in deep the type of training of the athletes. Where are differences between training session?
Statistical analysis
Please add more information about repeated-measures analysis of variance (ANOVA). Do you use the average mean of the variable? Please specify.
Moreover, I suggest reporting the F value in the manuscript.
Tables
I have some concerns about the results reported in the tables. Maybe I am misinterpreting the data.
If a comparison is significant, I do not understand why the average difference between the two groups [Mean difference (95% CI) ] is from a negative number to a positive number. To my knowledge, the value 0 should not be included. This also applies to Hedge's value (did you control for Bonferroni correction?).
Discussion
Please describe better the practical implication of your results. May be a distinction paragraph may help coaches and physical trainers to use our findings.
Author Response
This study aims to investigate weekly variations in the workload of Turkish national youth wrestlers during a season. I think that the article is not innovative in term of knowledge about the topic but adds information about internal training in wrestler population.
AUTHORS: DEAR REVIEWER, THANK YOU FOR YOUR INFORMATION.
Here my specific comments to the authors.
Lines 69-71
Why are there these differences? I suggest the authors to explain more in deep this point. How may the wrestling activity be affected by these differences?
AUTHORS: DEAR REVIEWER, THANK YOU. WE HAVE ADDED MORE INFORMATION.
Lines 78-80
I think that the relationship between acute/chronic workload ratio and injury is a lot discussed topic in the literature. I suggest of being more careful with this statement.
AUTHORS: DEAR REVIEWER, THANK YOU. WE HAVE REMOVED SUCH SENTENCE AND CHANGED.
Lines 100-110
I think that the authors should describe more in deep the type of training of the athletes. Where are differences between training session?
AUTHORS: DEAR REVIEWER, THANK YOU. WE HAVE ADDED GENERAL MACROCYCLE SAMPLE DURING PS.
Statistical analysis
Please add more information about repeated-measures analysis of variance (ANOVA). Do you use the average mean of the variable? Please specify.
Moreover, I suggest reporting the F value in the manuscript.
AUTHORS: DEAR REVIEWER, THANK YOU. WE ADD THESE INFORMATION THE STATICAL AND RESULT SECIONS.
Tables
I have some concerns about the results reported in the tables. Maybe I am misinterpreting the data.
If a comparison is significant, I do not understand why the average difference between the two groups [Mean difference (95% CI) ] is from a negative number to a positive number. To my knowledge, the value 0 should not be included. This also applies to Hedge's value (did you control for Bonferroni correction?).
AUTHORS: DEAR REVIEWER, INFORMATION WAS OBTAINED BY COMPARING THE DIFFERENCES BETWEEN THE NEXT AND PREVIOUS PERIODS.
Discussion
Please describe better the practical implication of your results. May be a distinction paragraph may help coaches and physical trainers to use our findings.
AUTHORS: DEAR REVIEWER, THANK YOU. WE HAVE ADDED A NEW PARAGRAPH.
Round 2
Reviewer 1 Report
First of all, I would like to thank the authors for their effort to respond to each comment. The manuscript needs a careful revision of the English. However, some new critical points have emerged. In particular, there are anomalies in the data reported. In particular, the strain calculation is incorrect (figure 2 a). If you examine the histogram graphs, you realize how the data do not match with other data reported in Figure 1a and Figure 2b. For example, the weekly training monotony and weekly acute load data of week 6 are characterized by low numbers (weekly training ∼800 AU, monotony ∼.05 AU), while the strain (∼1500 AU data is similar to that of the other weeks. Based on the formula reported by the Authors, the weekly strain for week 6 should be calculated as follows: 800 x 0.5= 400 AU. These anomalies are reported in Figure 2 and Table 3. I would suggest double-checking data before submission. Therefore, it becomes difficult to understand the results and part of the text must be rewritten in accordance.